# Predictors of cervical cancer screening uptake in two districts of Central Uganda

**Alone Isabirye** [1,2]*, **Martin Kayitale Mbonye**[1], **Betty Kwagala**[1]

**1** Department of Population Studies, School of Statistics and planning, College of Business and Management Sciences, Makerere University, Kampala, Uganda, **2** Department of Sociology and Social Administration, Faculty of Arts and Social Sciences, Kyambogo University, Kampala, Uganda

* aloneisab@gmail.com

**Data Availability Statement:** All relevant data are within the manuscript and Supporting Information files.

## Abstract

Uganda's cervical cancer age standardized incidence rate is four times the global estimate. Although Uganda's ministry of health recommends screening for women aged 25–49 years, the screening remains low even in the most developed region (Central Uganda) of the country. This study examined the demographic, social, and economic predictors of cervical cancer screening in Central Uganda with the aim of informing targeted interventions to improve screening. The cross-sectional survey was conducted in Wakiso and Nakasongola districts in Central Uganda. A total of 845 women age 25–49 years participated in the study. Data were analyzed at bivariate and multivariate levels to examine the predictors of CC (cervical cancer) screening. Only 1 in 5 women (20.6%) had ever screened for cervical cancer. Our multivariate logistic regression model indicated that wealth index, source of information, and knowledge about CC and CC screening were significantly associated with cervical cancer screening. The odds of cervical cancer screening were higher among rich women compared with poor women [AOR = 1.93 (95%CI: 1.06–3.42), p = 0.031)], receiving information from health providers compared with radios [AOR = 4.14 (95%CI: 2.65–6.48), p<0.001, and being more knowledgeable compared with being less knowledgeable about CC and CC screening [AOR = 2.46 (95%CI: 1.49–3.37), p<0.001)]. Overall cervical cancer screening uptake in central Uganda was found to be low. The findings of the study indicate that women from a wealthy background, who had been sensitized by health workers and with high knowledge about CC and CC screening had higher odds of having ever screened compared with their counterparts. Efforts to increase uptake of screening must address disparities in access to resources and knowledge.

## Background

Globally, cervical cancer ranks fourth amongst the most common types of cancer with about half and one third of a million new cases and deaths per annum respectively [1]. There is intense inequality in incidence globally since the biggest cervical cancer burden (84% of new cases and 87% of the deaths) occur in Low and Middle Income Countries (LMICs) [1]. Globally, Eastern Africa has the highest (30) and second highest (40.1) age standardized mortality

**Funding:** This work was funded by the Germany Academic Exchange Service (DAAD) (https://www. daad.de) funding program in-country/ in-region scholarship program Uganda 2016 under grant number 57299305 and awarded to IA. The funders had no role in study design, data collection and analysis, decision to publish, or preparation of the manuscript.

**Competing interests:** The authors have declared that no competing interests exist.

and incidence rates respectively [1]. While widespread cytology-based screening in high-income countries has resulted in decreased incidence and mortality from cervical cancer, LMICs, with poor uptake of screening, have not seen similar improvements, and in some cases, incidence and mortality actually continue to rise. Cervical cancer is the most common incident and mortal cancer amongst women in Uganda [2]. Cervical cancer contributes to about 40 percent of all malignancies reported by Kampala Cancer Registry (KCR) [3]. Estimates show that in 2018, about 6413 women were newly diagnosed with and 2400 succumbed to cervical cancer [2].

It is projected that by 2025 Uganda will have about 6400 new cervical cancer cases and 4300 deaths annually [2]. Services for cervical cancer prevention are relatively available in Central Uganda [4]; For instance, one of the study district was the first to benefit from a pilot project of cervical cancer prevention intervention [5]. It is also the most developed region of the country that hosts the capital city [6]. Studies have indicated high acceptability of CC screening in Central Uganda [7], especially self-collection of vaginal samples [8]. The Uganda's MOH target was to screen and treat 80% of eligible women aged 25–49 years by 2015 [4]. Ugandan women are screened by Visual Inspection with Acetic acid (VIA) and those who test positive with eligible precancerous lesions are treated by cryotherapy [4]. Screening in Uganda is unpredictable, opportunistic, and in some instances absent due to a shortage of resources or absence of will to commit financial resources [9]. This has resulted into low uptake [10] surprisingly, even in the most developed region (Central) of the country [11].

While comprehensive vaccination is cost-effective and lifesaving, incidence of cervical cancer is not expected to drop for at least 2 decades after widespread vaccination uptake [13] and in the meantime cervical dysplasia and early cancers will remain common and treatable through effective utilization of cervical cancer screening. The success of screening depends on access, uptake, and follow through the cascade to treatment for those who screen positive. According to World Health Organization (WHO), women aged 30 years should start screening for cervical cancer [14]. Additionally, the WHO guidelines recommend that screening at least once in a lifetime is beneficial, and intervals may depend on existing infrastructure and resources; decisions regarding the frequency of screening and target ages are determined by costs, existing burden of disease and infrastructure, and are left to respective governments [15].

Cytology-based screening is not practical for wide-spread use in sub-Saharan Africa due to its high cost, low sensitivity, inherent need for laboratories and trained technicians and complex follow-up protocols [4]. Testing for Human papillomavirus (HPV), the causative agent in almost all cervical cancer, is recommended as the primary screening modality where feasible [7,8]. HPV DNA testing is the most objective and sensitive screening approach [9–12], and has been shown to decrease mortality from cervical cancer in low-resource settings [13]. Visual inspection with acetic acid (VIA) is an acceptable alternative where HPV testing is cost-prohibitive [7,8]. Data suggest that self-collection of HPV, has comparable sensitivity to clinician-collection and is well-tolerated by women [11,12,14,15]. A simulation model based on epidemiologic data from Uganda shows that HPV testing may be more cost-effective than VIA [16]. Multiple proven cervical cancer screening approaches exist including: visual inspection with Lugol's iodine (VILI) or acetic acid (VIA), the Papanicolaou test (Pap smear), and HPV testing [16]. In resource rich settings, the WHO recommends testing for HPV first followed by VIA to identify women who can benefit from cryotherapy [15]. However, in resource limited settings, VIA is highly recommended because of its affordability and ability to screen and treat with in a single visit [16,17].

Previous studies about cervical cancer screening in Sub-Saharan Africa focused on either urban areas, health care settings or rural areas [10,18–20]. These studies have indicated high levels of awareness among study participants about cervical cancer, its signs, symptoms and

prevention [21,22], despite the low uptake of screening services. These studies further report several factors associated with cervical cancer screening, these include; age [23], social economic status [23], source of information [10], type of place of residence [24], knowledge about CC and CC screening [18,20,25]. Studies that used implementation science approach suggest that evidence on determinants of cervical cancer screening plays a significant role in informing effective interventions [26,27]. Interventions implemented with limited evidence regarding population specific predictors of cervical cancer prevention have faced challenges associated with lack of direction, negative perception, lack of scope, and limited acceptance, hence registering limited success [28,29]. Considering the scarcity of evidence on the predictors of cervical cancer screening with rural and urban representation, the objective of the current study was to report the prevalence of screening in Central Uganda and to examine for associations between predictors and successful screening in order to inform the design of future screening programs.

## Methods

### Study design and setting

We conducted a population-based cross-sectional survey in two of 27 districts in Central Uganda during June and July 2019. Central Uganda has 2 urban/ semi-urban districts and 25 predominantly rural districts. The two districts considered for the study include Wakiso; a peri-urban area near Kampala city and Nakasongola district; a rural district. Wakiso district is located at the outer skirts of the capital city (Kampala) with a total population of 1997418 people [30]. A number of activities in Wakiso are influenced by Kampala capital city. Nakasongola district is located 144 kilometers (kms) North of Kampala, with a total population of 181795 people [31], and the residents are mainly subsistence agriculturalists. According to 2014 National population and housing census area specific profiles, Wakiso and Nakasongola districts had 843604 and 31659 women aged 20–60 years respectively [30,31]. Cervical cancer prevention interventions have been implemented in these two districts [4]. The target population of the survey was women age 25–49 years who had lived in the area for at least six months. The 25–49 age group was considered because it is recommended by MOH for cervical cancer screening [4].

### Sample size and sampling procedure

The sample size of 850 women was calculated using Kish Leslie formula [32]. The prevalence of cervical cancer screening was estimated at 50% in order to obtain the maximum possible sample size which provided more precise estimates. The calculated study sample size of 850 was based on the estimated 50% prevalence and a precision of 5% to allow a 95% interval around estimates. Since the study used cluster sampling to select a simple random of clusters, we factored in the design effect of two and a response rate of 90%. One district was randomly selected from each of the rural (25 districts) and urban/ semi-urban (2 districts) clusters of central Uganda. We randomly selected 34 villages out of the 1916 villages from the study districts; 24 out of 1582 and 10 out of 334 villages/ wards were selected from Wakiso and Nakasongola districts respectively. A big proportion of eligible women were mainly from Wakiso therefore, we interviewed 40% more women from the district. Each selected village/ ward was considered a cluster and from each cluster, we selected 25 households using systematic sampling. From each selected household, one eligible woman was selected with priority given to spouses of household heads.

### Data collection procedure

We collected data using a structured pretested questionnaire containing items (questions) adapted from tools used in studies elsewhere [6,10,19,24]. For validation purposes, we

pretested the questionnaire with 10 women in the neighboring community (Mukono district); 10 kilometers away from the nearest study site to avoid contamination. The piloting of tools was conducted in an area that has characteristics similar to the study area. The tool was written in English and translation into Luganda language was done by two natives conversant in both English and the local language. We used a pair of translators who were not familiar with the original version of the questionnaire to back translate into English; the two versions were compared for conceptual equivalence and harmonized. A final translation into Luganda was then performed and checked for accuracy and preservation of meanings.

The survey tool consisted of seven sections. The first section included items on the women's demographics such as age, educational attainment, marital status, area of residence and previous health seeking behaviors. The second section contained questions on household factors such as type of house and household assets. The third section contained questions on reproduction such as number of children, contraceptive use. The fourth section had questions about CC, the fifth section had questions about cervical cancer screening, the sixth section had questions about HPV vaccination and the seventh section had husband's characteristics.

Interviews were conducted in Luganda, the major language spoken in Central Uganda. Our Research assistants (RAs) included five women from the study districts with Bachelor degrees in Social Sciences and Education. The RAs were trained for 2 days on principles of quantitative and survey research including data collection, the objectives of the research, and procedures for; sampling, interviewing and consenting. Each RA collected data from 6 to 8 participants per day for a period of 28 days. The data collected was reviewed by the Principal investigator on a daily basis to attain quality and comparability of data among RAs.

The outcome variable of the study was cervical cancer screening. Cervical cancer screening was measured in terms of whether respondents underwent any CC screening test ever; respondents were specifically asked "Have you ever been tested or examined for cervical cancer or precancer?"(No/Yes). Explanatory variables included; sociodemographic information including age, religion, place of residence, ethnicity, marital status, and parity; as well as health characteristics like use of family planning and recent visit to health facility. Other explanatory variables that were considered are; Source of information about CC screening, and distance to screening facility. Source of information about CC screening was obtained by asking women where they first got information concerning CC screening. Knowledge sections consisted of 11 and 8 items for CC and CC screening respectively. These questions examined the women's specific knowledge about CC and CC screening. One point was given if a respondent gave one or more correct response(s). We obtained composite knowledge and mean knowledge scores. Women who obtained scores greater than the mean were considered to have high knowledge and vice-versa. Wealth index was a composite score measured by household assets. Factor scores of household assets were generated. For this study it was recoded into three quintiles: poor, middle and rich.

## Data management and analysis

Two independent clerks entered data using Epidata 3.1 software (EpiData Software, Odense, Denmark). Data was synchronized and cleaned and then exported to STATA I/C version 16 for analysis [33]. Descriptive statistics in form of frequencies were generated and chi-squared tests were used to determine associations between independent variables including sociodemographic characteristics and dependent variables; being screened for CC ever. Multivariable logistic regression models were used to explore association of sociodemographic and health related predictors with the outcome (being ever screened for CC) adjusting to limit bias from confounding. Odds ratios were reported with accompanying 95% confidence intervals. The

multivariable logistic regression model comprised of explanatory variables whose p-values were less than 0.05 during the Chi-square tests. An exception was made for only type of residence because of its significance in the study. Multicollinearity tests were performed.

### Ethical considerations

This study was approved by Makerere University School of Social Sciences Research and Ethics Committee (MAHSSREC) and the Uganda National Council of Science and Technology (UNCST); UNCST registration number SS4848. We obtained clearance to access communities from the district and local leaders. Voluntary written informed consent was obtained from all participants, and they were assured of confidentiality. Participants were also informed of their freedom to decline participation if they chose to and or join the study and withdraw at any point without fear of retribution from the study team. After the interviews, about 5–10 minutes were allowed for participants to ask questions about cervical cancer and disease prevention.

## Results

All the 850 prospective participants approached accepted to participate in the study. Five questionnaires were incomplete (missing data on any three of residence, age, gender, number of biological children and level of education) and were excluded from analysis, leaving 845 for analysis. The study results indicate that only 1 in 5 (20.6%) of the women had ever screened for cervical cancer. Among the 845 women who participated in the survey majority were aged 25–39 years (80%), married (77.6%), in the middle wealth quintile (60.4%), Christians (80.2%), and had visited a trained health worker in the last 6 months (71.6%). Over half of the study participants were having 1–3 children (57.0%), using any form of contraception (52.5%), having high knowledge about CC and CC screening (58.3%), receiving information about cervical cancer screening from health workers (27.6%), and had attained at least secondary education (57.6%). Close to 4 in 10 (38.6%) were Baganda by tribe and very few (4.5%) women were professionals.

The bivariate results (Chi square results) indicate that cervical cancer screening was significantly associated with age (p = 0.001), occupation (p<0.001), wealth index (p = 0.020), knowledge about CC and CC screening (p<0.001), parity (p = 0.002) and source of information (p<0.001). Cervical cancer screening was higher among women who were; professionals (42.5%), rich (30.4%), having high level of knowledge about CC and CC screening (26.2%), having 4 or more children (25.6) and women whose main source of information about cervical cancer screening was health workers (41.5%). Our bivariate results indicate that religious affiliation, education attainment, study site, marital status, ethnicity, age at first marriage, visiting a health worker in the last six months, and use of contraception were not significantly associated with cervical cancer screening (Table 1).

### Associations between socio-demographic and economic factors with cervical cancer screening

We used multivariable logistic regression model to examine CC screening adjusting for study site, age, occupation, wealth index, distance to the screening facility, parity, and source of information about CC screening. Rich women [AOR = 1.93 (95%CI: 1.06–3.42), p = 0.031)] had 1.93 higher odds of having ever screened for cervical cancer compared to the poor women. Women whose main source of information about cervical cancer screening were health workers [AOR = 4.14 (95%CI: 2.65–6.48), p<0.001)] had 4.14 higher odds of having ever screened for the disease compared to women whose main source of information were radios. Women who had high knowledge about CC and CC screening [AOR = 2.46 (95%CI:

**Table 1. Distribution of women by demographics, socio-economic factors, knowledge levels and cervical cancer screening status (N = 845).**

| Characteristic | Ever screened n(%) | Never screened n(%) | Subtotal n(%) | P-value |
|---|---|---|---|---|
| **Total** | 174 (20.6) | 671 (79.4) | | |
| **Age group** | | | | 0.001 |
| 25–39 | 124 (18.3) | 555 (81.7) | 679 (80.4) | |
| 40–49 | 50 (30.1) | 116 (69.9) | 166 (19.6) | |
| **Religion** | | | | 0.349 |
| Christians | 144 (21.2) | 534 (78.7) | 678 (80.2) | |
| Muslims | 30 (18.0) | 137 (82.0) | 167 (19.8) | |
| **Study site** | | | | 0.918 |
| Wakiso (Urban) | 123 (20.5) | 477 (79.5) | 600 (71.0) | |
| Nakasongola (Rural) | 51 (20.8) | 194 (79.2) | 245 (29.0) | |
| **Education attainment** | | | | 0.694 |
| Some primary | 76 (21.2) | 282 (78.8) | 358 (42.4) | |
| At least some secondary | 98 (20.1) | 389 (79.9) | 487 (57.6) | |
| **Occupation** | | | | <0.001 |
| Professional | 17 (42.5) | 23 (57.5) | 40 (4.5) | |
| Other | 157 (19.5) | 648 (80.5) | 805 (95.5) | |
| **Marital status** | | | | 0.529 |
| Married | 132 (20.1) | 524 (79.9) | 656 (77.6) | |
| Single/ separated/ widowed | 42 (22.2) | 147 (77.8) | 189 (22.4) | |
| **Ethnicity** | | | | 0.818 |
| Baganda | 66 (20.3) | 260 (79.8) | 326 (38.6) | |
| Baluri | 42 (22.2) | 147 (77.8) | 189 (22.4) | |
| Others | 66 (20.0) | 264 (80.0) | 149 (39.1) | |
| **Age at first marriage** | | | | 0.702 |
| ≤18 | 67 (21.9) | 239 (78.1) | 306 (38.1) | |
| 19–34 | 100 (20.1) | 398 (79.9) | 498 (61.9) | |
| **Wealth index** | | | | 0.020 |
| Poor | 40 (17.9) | 183 (82.1) | 223 (26.4) | |
| Middle | 100 (19.6) | 410 (80.4) | 510 (60.4) | |
| Rich | 34 (30.4) | 78 (69.6) | 112 (13.3) | |
| **Visited a health worker in last six months** | | | | 0.519 |
| No | 46 (19.2) | 194 (80.8) | 240 (28.4) | |
| Yes | 128 (21.2) | 477 (78.8) | 605 (71.6) | |
| **Level of knowledge of CC and CC screening** | | | | <0.001 |
| Low | 45 (12.8) | 307 (87.2) | 352 (41.7) | |
| High | 129 (26.2) | 364 (73.8) | 493 (58.3) | |
| **Distance to screening facility (km)** | | | | 0.018 |
| ≤5 | 73 (23.8) | 234 (76.2) | 307 (36.3) | |
| 6–10 | 57 (23.0) | 191 (77.0) | 248 (29.4) | |
| ≥11 | 44 (15.2) | 246 (84.4) | 290 (34.3) | |
| **Currently using contraception** | | | | 0.475 |
| No | 78 (19.5) | 322 (80.5) | 400 (47.5) | |
| Yes | 93 (21.5) | 347 (78.5) | 445 (52.5) | |
| **Parity** | | | | 0.002 |
| ≤3 | 81 (16.8) | 401 (83.2) | 482 (57.0) | |
| ≥4 | 93 (25.6) | 270 (74.4) | 363 (43.0) | |

*(Continued)*

**Table 1.** (Continued)

| Characteristic | | | | |
| --- | --- | --- | --- | --- |
| | Ever screened n(%) | Never screened n(%) | Subtotal n(%) | P-value |
| **Source of information about screening** | | | | <0.001 |
| Radio | 41 (14.5) | 242 (85.5) | 283 (33.5) | |
| Health worker | 93 (41.5) | 131 (58.5) | 224 (26.5) | |
| Television | 17 (16.2) | 88 (83.8) | 105 (12.4) | |
| Others | 23 (9.9) | 210 (90.1) | 233 (27.6) | |

1.49–3.37), p<0.001)] had 2.46 higher odds of having ever screened compared to women with low knowledge (Table 2).

## Discussion

The results of our cross-sectional study reported low level of cervical cancer screening. In our study, we found that 1 in 5 (20.6%) of the women had ever screened for cervical cancer although the ministry of health's target was to screen 80% of the women aged 25–49 years by 2015 [4], four years before the survey. However, this finding from central Uganda is higher than findings from; Eastern Uganda (4.8%) [10], and Zimbabwe (9%) [24]. Though our findings are close to findings published from Tanzania (22.6%). Most of these studies had small sample size and were rural based. Age was not significant in predicting cervical cancer

**Table 2. Associations between socio-demographic and economic factors with cervical cancer screening.**

| Characteristic | AOR (95% CI) | P.value |
| --- | --- | --- |
| **District** | | |
| Nakasongola (Ref) | 1.0 | |
| Wakiso | 0.98 (0.63–1.53) | 0.927 |
| **Age group** | | |
| 25–39 (Ref) | 1.0 | |
| 40–49 | 1.49 (0.94–2.36) | 0.091 |
| **Wealth index** | | |
| Poor (Ref) | 1.0 | |
| Middle | 1.16 (0.74–1.84) | 0.520 |
| Rich | 1.90 (1.06–3.42) | **0.031**[*] |
| **Occupation** | | |
| Professionals (Ref) | 1.0 | |
| Others | 1.88 (0.89–3.99) | 0.098 |
| **Source of information** | | |
| Radio (Ref) | 1.0 | |
| Health worker | 4.14 (2.65–6.48) | **<0.001**[*] |
| Television | 0.98 (0.51–1.88) | 0.957 |
| Other | 0.79 (0.45–1.40) | 0.424 |
| **Parity** | | |
| ≤3 (Ref) | 1.0 | |
| ≥4 | 1.45 (0.97–2.18) | 0.069 |
| **Level of knowledge of CC and CC screening** | | |
| Low (Ref) | 1.0 | |
| High | 2.25 (1.49–3.37) | **<0.001**[*] |

screening when other explanatory variables were controlled for in a multivariable logistic regression model. The probable reason for this finding might be that screening behaviors are independent of age; with younger and older women attending screening on their personal initiative to remain healthy and health staffs' advise respectively [34]. However, this finding is contrary to studies from elsewhere [23,35,36] as they indicate a significant influence of age on cervical cancer screening.

Receiving relevant information regarding cervical cancer and cervical cancer screening recommendation from health providers has been found to positively affect the uptake of CC screening [10,11,36]. This finding is supported by our study which found that women who received relevant information from their health providers had higher odds of having ever screened. Health workers may be essential in health messaging because they are considered knowledgeable and trustworthy. Elsewhere, women who had discussions with health care providers regarding cervical cancer expressed higher intentions to screen [11,36]. The results of our study indicate that women's parity was not a significant predictor of cervical cancer screening. This is in agreement with findings from Kinshasa in the Democratic Republic of Congo [19] and Eastern Uganda [10]. Our finding is surprising because it is assumed that multiparous women have got higher probability of interaction with the health workers [6], these encourage those women to screen [11,36]. However, our findings are not supported by evidence from Nepal [35] and Jamaica [36] which indicated a positive influence of higher parity on cervical cancer screening. Our findings indicate that women's wealth index was positively associated with cervical cancer screening. This is consistent with prior studies [23,24,36]. The high prevalence of cervical cancer screening among women with a high wealth index may indicate their financial ability to afford screening services in a country where health insurance is limited [37].

Women with high knowledge about CC and CC screening had screened for cervical cancer compared to their counterparts. Several other studies have found the same result [18,38]. A study that integrated community health campaign with self-administered HPV screening in Kenya achieved high uptake [38]. Alternatively, it is likely that women become knowledgeable about the service as a result of seeking cervical cancer screening. We did not find a significant association between distance to screening facility and cervical cancer screening. This may point to the influence of other factors such as inability of women to pay for the service regardless of the distance. Only a quarter (25.5%) of the women in Uganda are in the highest wealth quintile [6] and the majority of public health facilities where these women would get the services free of charge are characterized by long waiting time, and few VIA providers [39]. Our finding is in support of prior evidence which found that proximity to services did not automatically translate into utilization [10,11]. However, our finding is not supported by findings of other studies [9,40]. These studies found a significant relationship between distance to services and utilization.

## Study limitations

We were not able to assess causation because of the cross-sectional nature of our study. Secondly, this study was done in mainly two districts in central Uganda. Consequently, the generalization of the study findings to other contextually different areas may be problematic. Finally, the study may have faced a problem of social desirability since the responses about cervical cancer screening were self-reported. However, probable bias was reduced by asking the women the duration since they last accessed the service. We selected the maximum possible sample size to have a fair representation of the women in the two districts of Central Uganda. Nevertheless, internal validity may have been affected by selection bias because women who were not found in their households and those who declined to participate were excluded.

## Conclusion

The findings of the study indicate a significant association between, wealth index, source of information, and high knowledge about CC and CC screening with cervical cancer screening. The above findings suggest that; provider-patient health education could be increased by utilizing times when reproductive age women are already interfacing with healthcare, like pregnancy since almost all women (97%) attend at least one antenatal visit in Uganda [6]. Screening opportunities should be expanded specifically to poor women. Alternatively, investment in interventions that increase women economic empowerment will increase the women's financial ability to afford health care.

## Supporting information

**S1 Dataset.**
(XLS)

**S1 File.**
(DOCX)

**S1 Questionnaire.**
(DOCX)

## Acknowledgments

The authors would like to acknowledge and thank the women who participated in this study. The authors also acknowledge the work of our research assistants.

## Author Contributions

**Conceptualization:** Alone Isabirye, Martin Kayitale Mbonye, Betty Kwagala.

**Data curation:** Alone Isabirye.

**Formal analysis:** Alone Isabirye.

**Funding acquisition:** Alone Isabirye.

**Investigation:** Alone Isabirye.

**Methodology:** Alone Isabirye, Martin Kayitale Mbonye, Betty Kwagala.

**Project administration:** Alone Isabirye, Betty Kwagala.

**Resources:** Alone Isabirye, Betty Kwagala.

**Software:** Alone Isabirye.

**Supervision:** Alone Isabirye, Martin Kayitale Mbonye, Betty Kwagala.

**Validation:** Alone Isabirye, Betty Kwagala.

**Visualization:** Alone Isabirye.

**Writing – original draft:** Alone Isabirye.

**Writing – review & editing:** Alone Isabirye, Martin Kayitale Mbonye, Betty Kwagala.

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
