## [Decision Letter · Decision Letter 0]

10 Jun 2020

PONE-D-20-02899

Predictors of Cervical cancer screening uptake in Central Uganda

PLOS ONE

Dear Dr. Isabirye,

Thank you for submitting your manuscript to PLOS ONE. After careful consideration, we feel that it has merit but does not fully meet PLOS ONE’s publication criteria as it currently stands. Therefore, we invite you to submit a revised version of the manuscript that addresses the points raised during the review process.

Your manuscript has been assessed by two reviewers, who raise partially overlapping concerns about the description of the study and methodology that should be addressed with the appropriate revisions. In addition, please ensure that you have discussed how (or whether) the questionnaire was pretested and validated and include a copy of the questionnaire as a supplementary information file. Finally, please specify what form of informed consent was obtained by study participants.

We look forward to receiving your revised manuscript.

Kind regards,

Emily Chenette

Staff Editor

PLOS ONE

Journal Requirements:

 2. In the Methods, please describe how the questionnaire was validated. If this did not occur, please provide the rationale for not validating the questionnaire.

- Please include additional information regarding the survey or questionnaire used in the study and ensure that you have provided sufficient details that others could replicate the analyses. For instance, if you developed a questionnaire as part of this study and it is not under a copyright more restrictive than CC-BY, please include a copy, in both the original language and English, as Supporting Information.

- In the ethics statement in the Methods and online submission information, please specify the type of informed consent that was obtained from the participants (for instance, written or verbal, and if verbal, how it was documented and witnessed).

- Please check and revise, as necessary, the p values reported in the manuscript as p cannot equal zero.

Reviewers' comments:

Reviewer's Responses to Questions

**Comments to the Author**

1. Is the manuscript technically sound, and do the data support the conclusions?

Reviewer #1: Partly

Reviewer #2: Partly

2. Has the statistical analysis been performed appropriately and rigorously? 

Reviewer #1: No

Reviewer #2: Yes

3. Have the authors made all data underlying the findings in their manuscript fully available?

Reviewer #1: Yes

Reviewer #2: Yes

4. Is the manuscript presented in an intelligible fashion and written in standard English?

Reviewer #1: Yes

Reviewer #2: Yes

5. Review Comments to the Author

Reviewer #1: Review notes

Abstract:

The Background section (lines 8-10), should state the problem (which is 2 fold: high burden cervical cancer, low rates of screening) and the objectives of this study, which I also see as two-fold: report prevalence of screening in these districts as well as investigate associations between various socio-demographic predictors and screening.

Methods: line 13 – pick either “univariate” or “bivariate” – they are the same.

Results: Line 15: age – is this relatively older or younger age? Line 15: source of what information? Health information? Line 15/16: the “knowledge” variable needs to be better defined throughout – I would re-define it here as “knowledge of importance of screening to prevent cervical cancer” or as an “understanding of the role of screening in cervical cancer prevention.” Similarly, the “logistics” variable needs to be better defined and I would re-define here as “knowing where to go for screening,” as you do in line 20. The odds ratios (lines 16-21) should be worked in with the prior sentence stating significant predictors. Please add “95%” prior to CI in lines 16-21. Also, please make clear if this is bivariate or multivariate analysis.

Conclusions: The conclusion needs to be re-written. Lines 23-24 only include some of the significant findings. Lines 25-30 go beyond the scope of the data. The first sentence is the overall finding and is good (line 22). I would limit the conclusion to 2-3 sentences in total. For the concluding sentence, I would offer something like: “relatively greater access to resources and knowledge, both disease-related and logistical, predicted a higher prevalence of screening” and then OK to have a final sentence with data-based recommendations for future. The last sentence could be something like: “efforts to increase uptake of screening must address disparities in access to resources and knowledge.”

Background:

In general, the first 2 paragraphs should be combined into 1 (or 2, but re-organized) paragraph(s) painting the picture of cervical cancer in LMICs and Uganda specifically, then you should have a paragraph about what is known about screening, then a description of this particular study’s goals.

Lines 33/34: make 1 sentence about ½ million new cases and 1/3 million deaths annually and cite Bray 2018. Take out the global age-standardized incidence rate as you discuss later (line 50) and it makes more sense there.

Lines 35/37: be careful about saying “disproportionately” as 85% of world’s population lives in LMICs, according to the World Bank, so 70% is not technically “disproportionate.” I think the 70% number is not accurate – The GLOBOCAN estimates more like 84% of new cases and 87% of the deaths occur in LMICs. You cite Torre 2012 as citation #2, but your citation for #1, the Bray GLOBOCAN estimates should replace the 2012 estimates. Take out citation 2 and update.

Line 37: Do you mean “sub”-Saharan Africa?

Lines 38-40: Take out the India statistic and it’s confusing to also hear about breast cancer – I would just focus on the incidence and mortality rates, for which Southern and Eastern African regions are the highest in the world. Be careful: your references #3 and #4 are the same and I’m not sure are complete.

Instead of 41-45: I would end this paragraph by saying something like: “while widespread cytology-based screening in high-income countries has resulted in decreased incidence and mortality from cervical cancer, LMICs, with poor uptake of screening, have not seen similar improvements and, in some cases, incidence and mortality actually continue to rise.” If you want to include the WHO recommendations, I would instead use the WHO guidelines for screening and treatment of precancerous lesions for cervical cancer prevention, 2013. Your statement in line 42 about screening (do you mean once per lifetime screening recommended for 25-49yos?) is an over-simplification of the recommendations. Not sure the SDGs add and reference #6 seems to be incomplete.

Lines 47-64: this paragraph is all Uganda-specific. It’s fine to follow the first general paragraph about high prevalence cervical cancer and low prevalence screening in LMICs, but this paragraph needs to be shorter and more focused:

- Lines 47-49: summarize that Cervical cancer is most common incident and mortal cancer among women in Uganda

- Lines 49-51: one sentence with age-standardized incidence and mortality rates in UG compared to global OK to underscore the magnitude of the problem. Be careful of your citation, I would use the Bruni HPV Uganda-specific report for the citation – make sure the citation is complete (For example, I have cited as: Bruni L B-RL, Albero G, Aldea M, Serrano B, Valencia S, Brotons M, Mena M, Cosano R, Muñoz J, Bosch FX, de Sanjosé S, Castellsagué X. Human Papillomavirus and Related Diseases in Uganda. Summary Report 2016- 02-26: ICO Information Centre on HPV and Cancer (HPV Information Centre), 2016.)

- Lines 52-56 can essentially be cut as long as you have the summary statement about cervical cancer as the most common incident and mortal cancer among women in uganda.

- Line 56-57: cut this sentence about 8/10 women at UCI have cervix cancer. Reference 9: seems incomplete the way it is cited and probably OK to just cite the GLOBOCAN 2018 instead of the Uganda Fact Sheet.

- Lines 57-60: I’m not sure what these lines add to the intro. This is not a paper about cancer, it’s a paper about screening, so I think sufficient to make a grim summary statement about cervical cancer earlier on. I would cut these sentences and references 10-12 (also, these references seem incomplete).

- Lines 60-64: this is a good way to end this paragraph, to bring it back to Uganda. Lines 60-63 should be one sentence about the MOH goal to screen 80% - be careful, your citation for #13 is incomplete: change to “MOH. Strategic Plan for Cervical Cancer Prevention and Control in Uganda, 2010-2014. Kampala, Uganda: Ministry of Health; 2010.” Line 63-64 – what study are you citing here? Citation 14 is incomplete? It might be stronger to start the next paragraph with the best estimate of screening prevalence in Uganda.

Lines 66-85: This paragraph is too long and is disorganized. It is a good idea to use this paragraph to discuss what it is we know about screening and why we need to know more in order to improve screening programs. Some of what is written in this paragraph should be removed and saved for the discussion where you will compare this study’s findings to that of other studies.

- The paragraph could start with your statement about how research on determinants play a role in designing effective interventions (lines 69-70), but its not clear to me that your references 17-21 (careful, most of these references are incomplete and be consistent with either numbering) actually justify this claim – are you suggesting that these studies used some sort of implementation science approach to designing their interventions?

- Lines 72-73 say another general statement about how cervical cancer is preventable if screening is implemented – your citations 22 and 23 don’t really justify this statement which is generally accepted to be true, and the citations are incomplete. This general statement does not belong at this point in the introduction.

- Lines 74-76 and 66-68 go together and given an overview of what is known about cervical cancer screening – be careful of your references, some are incomplete. I would stick to references about Eastern Africa.

- Lines 76-81 focus on urban vs rural – the first sentence is incomplete and unnecessary. The next 2 sentences are too long are not focused (Lines 77-81) – are you just trying to say that living in a rural area is predictive of not screening?

- Lines 81-85 talk about “strengths” of the current study – this goes in discussion, not background. Instead, end the paragraph with a statement about the objective of the current study. Something like, “the objective of the current study was to report the prevalence of screening in Central Uganda and to examine for associations between predictors and successful screening in order to inform the design of future screening programs…”

In general, this is a paper about screening and I am surprised that I don’t find anything in the introduction about VIA versus HPV vs cytology for methods screening. Generally speaking, there is too much about cervical cancer and not enough about screening.

Methods:

The first section should include not only a description of study setting, but also study population – you can include the age-range here… other inclusion/exclusion? Language?

Line 95: reference 34 is incomplete.

Lines 95-97: you claim there is no precedent in the literature, but in line 110, you cite studies which likely report on a prevalence of screening. There are estimates out there – if you think they are underestimates (50% is high), you need to explain why you think prior studies would be underestimate and so you’ve gone with 50%.

Lines 97-98, I think you have overestimated your sample size. You could say something like, “To calculate our sample size, we assumed that if 50% of women in Central Uganda had previously screened for cervical cancer, a total of 402 women would allow us to experimentally determine this proportion within 5% (confidence level 95%) using a binomial “exact” calculation (Hulley SB, Cummings SR, Browner WS, Grady D, Newman TB. Designing clinical research : an epidemiologic approach. 4th ed. Philadelphia, PA: Lippincott Williams & Wilkins; 2013. Appendix 6E, page 81). https://www.sample-size.net/sample-size-conf-interval-proportion/

Lines 98-100: this is not a “cluster randomized study,” rather, you used “cluster sampling” to select a simple random sample of clusters. This is very different and needs to be clarified.

Lines 99-101: Why 70% from Wakiso and 30% from Nakasongola? Explain and provide justification.

Line 101: “the target population…” – this belongs in “inclusion criteria.” Reference 35 does not make sense here. If you are justifying the ages in your inclusion criteria, I would cite Ugandan MOH recommendations for screening a particular age range (they recommend 25-49).

Lines 101-103 should be removed here – this is for inclusion criteria. For sampling procedure, describe the cluster sampling technique.

Lines 109-110: Were any of the questionnaires validated? Doesn’t make sense to say that the questionnaire was adapted from “findings” from other studies – sounds like it was modified from questionnaires used in other studies.

Lines 119-124: should be a separate paragraph.

Lines 124-127: the description of the research assistants and their training could be a separate paragraph.

Lines 127-128: talk more about the pre-test/pilot phase – were questions adapted? Community participation is an important part of study design and this could be highlighted here.

Lines 128-129: remove the sentence “the questionnaires were administered by 5…” – this is redundant.

Lines 129-131: this is a good way to summarize, but also, were data electronically captured? Sent to a secure server?

Line 134: a cervical examination is not screening – do you mean VIA? This has to be better defined.

Line 135-136: screening does not test for cervical cancer, but also pre-cancer/dysplasia…

Lines 133-161: this paragraph is way too long and needs to be split up. You don’t need to include so much information about basic demographic variables, just need to say that demographic info including age, religion, place residcen, ethnicity, marital characteristics, parity; as well as health characteristics like use fam planning, recent visit to health center. Lines 147-148: this is the first I’m hearing about this “autonomy” variable – I did not see in abstract, if not used in analysis, don’t need to describe. The description of the “knowledge” variable 149-157 is too long and should be truncated/summarized. Line 157-158 – the justification doesn’t need to be there and the reference 39 is incomplete and doesn’t make sense. Lines 158-161 – description wealth index is too long and also was presented in abstract as dichotomous (rich/poor).

There is description of the “source of information” variable – as that is not an obvious variable, please explain – source of what information, exactly?

Line 164: I don’t understand how Epidata was used, clarify.

Lines 166-171 need to be re-written. I image you used chi-square tests to test for association between outcome (being screened ever) and categorical explatory variables. And I imaging that for continuous explanatory rpedictors, you used t-tests to compare sample means by outcome. I imagine you used logistic regression to explore association demographic and health-related predictors with outcome, adjusting to limit bias from confounding. Discuss how you build the multivariate model and whether sensitivity analyses were performed.

Line 168: you sat that knowledge is a dependent variable, yet you treat it an an independent predictor and your dependent variable is your outcome variable, screening. Fix this.

Results

The results section needs to be rewritten. The section needs to lead with the proportion who had ever been screened (your main outcome).

Tables 1 and 2 should be combined into one table, I would suggest a table that looks like this:

Characteristic Total (%) History prior screening (%) Never been screened (%) OR unadjusted prior screening (95% CI) OR adjusted prior screening (95% CI)

Total (N, %) 845 XXX (21%) XXX (79%)

Age

<=29 315 (37%) 13 87 1.0 1.0

30-39 364 (43%) 23 77 1.4 (0.83-2.19) x.x (a-b)

40-49 166 (20%) 30 70 1.9 (1.05-3.55) x.x (a-b)

Some specific feedback:

Line 192: remove sentence “results in table one…”

Lines 192-200 – the order of the variables needs to be re-organized. For example – Catholics are mentioned in 196 and then protestants are mentioned in line 199. You are describing the population in this paragraph, which is a fine was to start.

Line 202-203 : remove this first sentence. Lines 203- 212: this needs to be made into one paragraph and re-organized. Also, are you discussing bivariate or multivariate analysis? What is going on with the “knowledge” variable? In the Methods section, you discuss a composite score variable, but here there are two knowledge variables – please be consistent.

Lines 223-225 – remove this sentence.

Lines 227-235 – need to be combined and re-written with lines 192-203. This is all the same information.

Some other thoughts about the variables (in table one):

- Age – do you need three categories? Often easier to understand as a dichotomous variable

- Religion – consider combining as Christian versus Muslim

- Study site: note which is urban/which is rural

- Education: consider a dichotomous variable – completed primary (at least some secondary) versus not completed (some primary)

- Occupation: consider professional versus other (“proffessional" is spelled wrong)

- Marital status: consider married vs single/widow/separated

- Ethnicity: consider combining for fewer categories

- Age at first marriage: the “single” category should not be in there.

- Knowledge: this is different than described in Methods and is different table one to table two. Make consistent

- Distance to screening facility: “don’t know where to go” does NOT belong in there – that is a different variable… is that different than “awareness of screening services?”

Discussion

Overall, this is the best-written section of the paper, but it still needs significant work.

Lines 240-246: good to start with your overall finding about screening prevalence. I would take out the first sentence. I don’t think it’s necessarily useful to include stats from Nepal, I would stick to the region.

Lines 248-252: this is about knowing where to go – the logistics variable. I would remove the words “the possible explanation for this finding is that” and just start the sentence with “Increased awareness of …”

Lines 254-259 are about the age variable. This paragraph is OK.

Lines 260-269: this paragraph is about he “source of information” variable. As I’ve mentioned before, this variable needs to be better defined. I would remove the sentence lines 263-265 “this observation might be due to..” I would also remove the sentence lines 267-269 “Secondly it…) I would add a sentence positing that health workers may be essential in health messaging.

Lines 271-274, this paragraph is about the wealth variable – I would re-write the first sentence – take out “corrobortating the results…” part. Overall the paragraph is ok.

Lines 276-281, the “knowledge” variable is problematic and I have brought that up before. I would take out he sentence 278-279 “understanding the impotance…” – this is speculation. Your last sentence is an important point that prior screening is likely related to increased knowledge.

Lines 284-286: Study limitations: can’t assess causality and generalizability. What about sources of bias?

You have two nearly identical paragraphs: recommendations and conclusions – this should just be there once, likely as “conclusions”

Lines 289-295: take out “substantial effect” line 290 and change to something like “associated” – you cannot imply causation. You then focus on the source of information variable and suggest that antenatal providers should discuss screening – this is a really important point and should be developed! Why not screen in pregnancy? What proportion of women interact with providers during pregnancy (at least one anenatal visit)? This also brings in the knowledge variable – be explicit about how these are related. You then focus on the wealth index variable and suggest universal healthcare – what are some challenges there? Does the government have the funding? What would make cervical cancer screening cost effective?

References (lines 317-423) – nearly all the references are incomplete – please be careful and re-do the references.

Reviewer #2: REVIEWER’S REPORT

MANUSCRIPT NUMBER: PONE-D-20-02899

MANUSCRIPT FULL TITLE: Predictors of Cervical cancer screening uptake in Central Uganda

GENERAL COMMENTS

What are the main claims of the paper and how significant are they for the discipline?

Isabirye et al., indicated that they had identified and determined the extent of the association of predictors with screening for cervical cancer among women in two districts of Uganda. These finding provide evidences for the importance of country/community-specific information for the promotion of cervical cancer screening.

Are the claims properly placed in the context of the previous literature? Have the authors treated the literature fairly?

The background is overload with information on cancer distribution (over 60% of the text) while the main focus of the study, which is predictors of screening for cervical cancer are less than 20%. Furthermore, there was nothing about screening, the authors should note that the types of screening available in a community has been show to also be a predictor of participation in screening, particularly in the context of cultural limitation. Again, the problems or concerns associated with the lack of knowledge of the predictors of cervical cancer screening in Uganda have not been discussed at all. Therefore, I suggest the background be written again with much focus on the key words of the objective of the study

Do the data and analyses fully support the claims? If not, what other evidence is required?

The claims are largely supported by the data and analysis, however, the table 1 should be reported in a more informative manner as suggested in the specific comments below. I was wondering why there was no analysis comparing urban to rural women, since the selection of the districts ensure this representation. Was that the difference was not significant?

PLOS ONE encourages authors to publish detailed protocols and algorithms as supporting information online. Do any particular methods used in the manuscript warrant such treatment?

No method used in this study warrants a publication of detailed protocol as supporting information online

If a protocol is already provided, for example for a randomized controlled trial, are there any important deviations from it? If so, have the authors explained adequately why the deviations occurred?

None of such protocols was identified in this manuscript

If the paper is considered unsuitable for publication in its present form, does the study itself show sufficient potential that the authors should be encouraged to resubmit a revised version?

Yes, the authors are encouraged to make the suggested revisions

Are original data deposited in appropriate repositories and accession/version numbers provided for genes, proteins, mutants, diseases, etc.?

The Manuscript gives the indication that all data are fully available without restriction upon reasonable request from the corresponding author.

Are details of the methodology sufficient to allow the experiments to be reproduced?

Details of the method for the selection of the 2 of the 27 districts, how the number of villages selected from each district was arrived at, and the selection of women within each village need to be provided.

Is the manuscript well organized and written clearly enough to be accessible to non-specialists?

The manuscript is well organized and clear

SPECIFIC COMMENTS

LINE 1: The title should read as “Predictors of cervical cancer screening uptake in two districts of Central Uganda”

This because the 2 districts have not been shown to be representative of the 27 district from which they were selected, therefore, the data should not be generalized to Central Uganda. It does not have the claimed external validity.

LINE 13: It will be more appropriate to rather state the type of statistic used to analyse what variable or for what purpose or which outcome.

Lines 29-30: No data/finding has been provided to support this conclusion, this should not be a conclusion of this study.

LINE 37: You mean "Sub-Saharan Africa region"

Lines 41-45: These statement are hanging. Authors need to the question "and so what?" in respect to this statement.

Lines 51: “25” should be “25.0”, and “Global” should be “global”

Lines 81-82: This should state clearly the overall aim of the study, not just a part of it.

Line 83: how was this minimisation achieved?

Lines 90-92: How were the two districts selected from the 27 districts? How many of the 27 districts were rural and urban/peri-urban? Provide more information regarding this selection for the reader to be convinces the selection was representative and fair.

Lines 98-99: the study was earlier on described as a cross-sectional study, so why call it a cluster random study here? Furthermore, there is no description of how the two districts were selected, and no grouping (clusters) have been described.

Lines 99-101: what informed this proportions?

Line 102: Please recheck this fact and determine if it applies to all countries?

Lines 104-105: Why 24 and 10 villages? How were the women recruited from the villages?

Line 128: Were these "nearby community" near by the study area or the place the study was designed? Please state the names and how far they are from the study districts or villages. This is to convince the reader that there was a very low probability of contamination of the study villages.

Line 128-130: Is this per village or per day?

If each of the 5 RAs collected data from 6-8 participants in each village and 34 villages were involved in this study then 1020 - 1360 participants were involved. However, if each of the 5 RAs collected data from 6-8 participants per day and 28 day were used for data collection then 840 - 1120 participants were involved, which accounts for only 850 participants in the study?

Line 178: Was the voluntary informed consent written or oral?

Line 193: was this proportion not designed to be so as indicated in the methods? If not, that is this was observed after recruitment, then it should be taken out of the methods. By the way it was reported as 70% in the method.

Lines 192-200: some of the numbers do not have the “%”

Line 218 (Table 1): Table should be improved. I suggest a proper cross-table as indicated below. All p value stated as “0.000” should be stated as “<0.0001”

Screening, n (%) Chi-squared p-value

Ever screened Never screened Subtotal (%)

Age group 0.001

< 29 n1 (%) n2 (%) n1+ n2(%)

30-39

40-49

Total

Line 225: Were all variables used in the multivariable logistic analysis? How were they entered into the multivariable analysis?

Lines 244-246: These seem inappropriate, comparing a region to a hospital what is the purpose of the comparison? Were the studies similar to this study, if not indicate the differences.

Lines 255-256: Rather, state what the other similar studies found, what this study found is already stated in the results.

Line 268: Please provide a reference.

Lines 289-290: I suggest this is deleted, it is not a recommendation.

Lines 293-295: there is no finding in this study that relates to this recommendation. who are the recommended being directed at?

Lines 297-298: No need to repeat the results in the conclusion, Delete “Only 1 in 5 women (21%) had ever screened for cervical cancer.”

Lines 300-305: how different is this from a recommendation, as stated earlier? Please delete.

Line 350: check reference, seems incomplete

Line 351: check numbers in author name

Line 365: check et al

Line 398: upper case

Line 413: check reference, seems incomplete

6. PLOS authors have the option to publish the peer review history of their article (what does this mean?). If published, this will include your full peer review and any attached files.

Reviewer #1: Yes: Megan Swanson

Reviewer #2: Yes: Adolf Kofi Awua

---

## [Author Response · Author response to Decision Letter 0]

16 Aug 2020

We appreciate your critical reflection on our scientific piece. Thank you.

---

## [Decision Letter · Decision Letter 1]

11 Nov 2020

PONE-D-20-02899R1

Predictors of cervical cancer screening uptake in two districts of Central Uganda

PLOS ONE

Dear Dr. Isabirye,

Thank you for submitting your manuscript to PLOS ONE. After careful consideration, we feel that it has merit but does not fully meet PLOS ONE’s publication criteria as it currently stands. Therefore, we invite you to submit a revised version of the manuscript that addresses the points raised during the review process.

Please respond to additional comments by a reviewer.

We look forward to receiving your revised manuscript.

Kind regards,

Clement A. Adebamowo, BM, ChB Hons; FWACS, FACS, ScD, FASCO

Academic Editor

PLOS ONE

Reviewers' comments:

Reviewer's Responses to Questions

**Comments to the Author**

1. If the authors have adequately addressed your comments raised in a previous round of review and you feel that this manuscript is now acceptable for publication, you may indicate that here to bypass the “Comments to the Author” section, enter your conflict of interest statement in the “Confidential to Editor” section, and submit your "Accept" recommendation.

Reviewer #1: All comments have been addressed

Reviewer #2: All comments have been addressed

2. Is the manuscript technically sound, and do the data support the conclusions?

Reviewer #1: Yes

Reviewer #2: Yes

3. Has the statistical analysis been performed appropriately and rigorously? 

Reviewer #1: Yes

Reviewer #2: Yes

4. Have the authors made all data underlying the findings in their manuscript fully available?

Reviewer #1: Yes

Reviewer #2: Yes

5. Is the manuscript presented in an intelligible fashion and written in standard English?

Reviewer #1: Yes

Reviewer #2: Yes

6. Review Comments to the Author

Reviewer #1: Reviewer Notes

Predictors of cervical cancer screening uptake in two districts of Central Uganda

Overall, this is a much stronger version. Thank you for working to revise the manuscript. The following are minor suggestions for revision.

Abstract:

- I think you can remove lines 13-14

- Line 16, after “targeted interventions”, I would add “to improve screening”

- Line 26, as you are shifting to conclusion, I would add the word “Overall,” cervical cancer screening low…

- Lines 29 and 30: remove “the above findings suggest that” and start this sentence with “Efforts to increase uptake…”

Background: In general, this is much improved with the emphasis on screening, rather than cancer, but I still think the section could be shortened.

- I would take out some of the cancer-specific stats from paragraph one and transition quicker to the problem of inadequate screening. Consider splitting into two paragraphs.

- Line 57: need to write-out “Visual inspection with acetic acid” before VIA

- Line 64: You already defined LMICs

- Lines 63-67: consider removing these lines and starting with “the success of screening depends on uptake…” – also depends on follow-through the cascade to treatment. Vaccination is beyond the scope of your discussion. If you are going to mention vaccination, I would say something to the effect of “while comprehensive vaccination is cost-effective and lifesaving, incidence of cervical cancer not expected to drop for at least 2 decades after widespread vaccination uptake (Jit M, Brisson M, Portnoy A, Hutubessy R. Cost-effectiveness of female human papillomavirus vaccination in 179 countries: a PRIME modelling study. The Lancet Global health. 2014) and in the meantime cervical dysplasia and early cancers will remain common and treatable”

- Lines 72-79: Be careful, comparing various screening methods is beyond the scope of this paper – the debate between HPV testing and VIA is nuanced and there are strong feelings on both sides.

Methods:

- Line 181-182: in forward-stepwise model building, usually cast a wider net and include predictors with p value more like 0.2 or less and also those known to be associated a priori. Just a comment.

- Lines 183-184, shorten to: “multicollinearity tests were performed”

Results:

- Lines 201 – 207 – you can just refer to table for characterization of the demographics.

- I had previously suggested combining tables 1 and 2. OK to have separate tables, but I would recommend using bivariate ORs, rather than chi-square tests.

- Lines 226-232 – careful – it’s not 1.93x more likely, but, rather, higher odds – this error is repeated.

- Table 2: add 1.0 for the OR for the referent groups. Personally, I don’t think you need p-values if you are giving CIs.

Discussion:

- line 239: you said “1 in 5” before and now saying “2 in 10”

- line 255: type “messaging”

Conclusion:

- lines 297-299: careful about the antenatal screening recommendation - ASCO specifically recommends NOT screening in pregnancy (I disagree with the recommendation and write an opinion piece about it which was published in JGO) – I would take a step back here and suggest that one approach to increasing the provider-pt health education could be to utilize times when reproductive age women are already interfacing with healthcare, like pregnancy.

- Be careful talking about health insurance coverage in Uganda – tie the sentiment to your findings, that “rich” women were more likely to get screened – thus, your recommendation is for screening opportunities to be expanded specifically for poor women – If you are going to talk about national financing for the health sector, this is a huge topic and I would likely rather just hint at it here.

- Seems like your main findings were that rich women, women who learn from providers and women with high knowledge were more likely to screen. But seems like the conclusion only really touches on the first 2 – do you want to somehow also bring in the knowledge variable here?

Reviewer #2: (No Response)

7. PLOS authors have the option to publish the peer review history of their article (what does this mean?). If published, this will include your full peer review and any attached files.

Reviewer #1: **Yes: **Megan L Swanson

Reviewer #2: **Yes: **Dr Adolf Kofi Awua

---

## [Author Response · Author response to Decision Letter 1]

17 Nov 2020

Response letter

Date 15 November 2020

To: PLOS ONE <plosone@plos.org

From: "Alone Isabirye" aloneisab@gmail.com

Subject: Response to review comments of our manuscript submitted to PLOS ONE (PONE-D-20-02899R1)

PONE-D-20-02899R1

Predictors of Cervical cancer screening uptake in two districts of Central Uganda

Alone Isabirye, Martin Kayitale Mbonye, Betty Kwagala

PLOS ONE

Dear Editor,

Thank you for your reply regarding our manuscript "Predictors of Cervical cancer screening uptake in two districts of Central Uganda" (PONE-D-20-02899R1). We are grateful for the reviewers’ comments. We have revised and modified the manuscripts according to the referees’ critiques. As a consequence, we provide a revised manuscript with the reviewers’ suggestions integrated therein:

Response to editor’s comments

-Deposit laboratory protocols in protocols.io; Not applicable

-Uploading our figure files to the Preflight Analysis and Conversion Engine (PACE) digital diagnostic tool; Not applicable

Response to reviewer 1’s comments

The overall comment about the revised manuscript is appreciated. 

Abstract:

- Lines 13-14 were removed.

- Line 16, we added “to improve screening” after “targeted interventions” 

- Line 26, as we are shifting to conclusion, we added the word “Overall,” cervical cancer screening low…

- Lines 29 and 30: we removed “the above findings suggest that” and started the sentence with “Efforts to increase uptake…”

Background: 

- We removed some cancer-specific statistics from paragraph one especially about Uganda and the Cancer Institute and transitioned quicker to the problem of inadequate screening. Additionally, the paragraph was split into two paragraphs.

- Line 57: we wrote-out “Visual inspection with acetic acid” before VIA

- Line 64: second definition of “LMICs” was removed

- Lines 63-67: The section was revised while considering the proposed recommendations by the reviewer.

- Lines 72-79: Phrase comparing various screening methods was removed. 

Methods: 

- Line 181-182: The comment of casting a wide net in variable consideration is appreciated. We included some variables known to be associated a priori like type of residence. However, almost all variables left out in our multivariate model had p-values far higher than 0.2.

- Lines 183-184, the statement was shortened to: “multicollinearity tests were performed”

Results:

- Lines 201 – 207 – The table was referred to. 

- We are pleased with your decision to accept the two tables. We also humbly propose that chi-square test is used at bi-variate level. This is because data was analyzed at three levels and the selection of statistical approaches at the respective levels was guided by Bloom’s taxonomy of learning objectives.

- Lines 226-232 – 1.93x more likely was corrected to “higher odds” – this error was fixed throughout the manuscript.

- Table 2: 1.0 was added for the OR for the referent groups. The authors preferred presenting the results with both p-values and confidence intervals because these statistical measures complement one another.

Discussion:

- Line 239: stating “1 in 5” before and stating “2 in 10” later was systematized by mentioning “1 in 5” throughout.

- line 255: We correctly typed “messaging” 

Conclusion:

- Lines 297-299: The antenatal screening recommendation was fixed as suggested by the reviewer. 

- Our recommendation regarding poor women was fixed as proposed by the reviewer.

- We included high knowledge among our main findings.

Response to reviewer 2’s comments

- No comments.

We hope that our modifications render our manuscript in its current form suitable for publication in PLOSONE

Yours sincerely,

Isabirye Alone

aloneisab@gmail.com

On behalf of the authors

---

## [Editor Report · Decision Letter 2]

19 Nov 2020

Predictors of cervical cancer screening uptake in two districts of Central Uganda

PONE-D-20-02899R2

Dear Dr. Isabirye,

We’re pleased to inform you that your manuscript has been judged scientifically suitable for publication and will be formally accepted for publication once it meets all outstanding technical requirements.

Kind regards,

Clement A. Adebamowo, BM, ChB Hons; FWACS, FACS, ScD, FASCO

Academic Editor

PLOS ONE
---

## [Editor Report · Acceptance letter]

23 Nov 2020

PONE-D-20-02899R2 

Predictors of cervical cancer screening uptake in two districts of Central Uganda 

Dear Dr. Isabirye:

I'm pleased to inform you that your manuscript has been deemed suitable for publication in PLOS ONE. Congratulations! Your manuscript is now with our production department. 

Kind regards, 

on behalf of

Dr. Clement A. Adebamowo 

Academic Editor

PLOS ONE